# C/EBPβ Regulates TFAM Expression, Mitochondrial Function and Autophagy in Cellular Models of Parkinson’s Disease

**DOI:** 10.3390/ijms24021459

**Published:** 2023-01-11

**Authors:** Ana Sierra-Magro, Fernando Bartolome, David Lozano-Muñoz, Jesús Alarcón-Gil, Elena Gine, Marina Sanz-SanCristobal, Sandra Alonso-Gil, Marta Cortes-Canteli, Eva Carro, Ana Pérez-Castillo, José A. Morales-García

**Affiliations:** 1Instituto de Investigaciones Biomédicas A. Sols (CSIC-UAM), Arturo Duperier, 4, 28029 Madrid, Spain; 2Networking Research Center on Neurodegenerative Diseases (CIBERNED), Instituto de Salud Carlos III, 28031 Madrid, Spain; 3Group of Neurodegenerative Diseases, Hospital Universitario 12 de Octubre Research Institute (imas12), 28041 Madrid, Spain; 4Department of Cellular Biology, School of Medicine, Universidad Complutense de Madrid, 28040 Madrid, Spain; 5Centro Nacional de Investigaciones Cardiovasculares (CNIC), Melchor Fernández Almagro, 3, 28029 Madrid, Spain; 6Instituto de Investigación Sanitaria Fundación Jimenez Díaz, Avenida de los Reyes Católicos, 2, 28040 Madrid, Spain

**Keywords:** C/EBPβ, TFAM, mitochondria, autophagy, Parkinson

## Abstract

Parkinson’s disease (PD) is a neurodegenerative disorder that results from the degeneration of dopaminergic neurons in the *substantia nigra pars compacta* (*SNpc*). Since there are only symptomatic treatments available, new cellular and molecular targets involved in the onset and progression of this disease are needed to develop effective treatments. CCAAT/Enhancer Binding Protein β (C/EBPβ) transcription factor levels are altered in patients with a variety of neurodegenerative diseases, suggesting that it may be a good therapeutic target for the treatment of PD. A list of genes involved in PD that can be regulated by C/EBPβ was generated by the combination of genetic and in silico data, the mitochondrial transcription factor A (TFAM) being among them. In this paper, we observed that C/EBPβ overexpression increased TFAM promoter activity. However, downregulation of C/EBPβ in different PD/neuroinflammation cellular models produced an increase in TFAM levels, together with other mitochondrial markers. This led us to propose an accumulation of non-functional mitochondria possibly due to the alteration of their autophagic degradation in the absence of C/EBPβ. Then, we concluded that C/EBPβ is not only involved in harmful processes occurring in PD, such as inflammation, but is also implicated in mitochondrial function and autophagy in PD-like conditions.

## 1. Introduction

Parkinson’s disease (PD) is the second most prevalent neurodegenerative disorder affecting the elderly after Alzheimer’s disease (AD). It is characterized by the loss of dopaminergic neurons in a specific region of the ventral midbrain called *substantia nigra pars compacta* (*SNpc*), which causes motor symptoms such as bradykinesia, akinesia, resting tremor, rigidity, or postural abnormalities; and non-motor symptoms, such as dementia, hyposmia, or depression [1]. The presence of protein aggregates that are called Lewy bodies can be observed in PD patients’ brains, with their main protein component being α-synuclein (α-syn) [2]. Global life expectancy is increasing, leading to an increase in PD prevalence that has an enormous economic and human cost [3,4]. To date, there is no treatment that forestalls the progressive degeneration of dopaminergic neurons, revealing the necessity of finding new molecular targets for the treatment of PD [5]. Several processes have been implicated in PD pathogenesis, such as alterations in the endosomal–proteasomal–autophagy pathway, the increase in misfolded protein aggregation, oxidative stress, mitochondrial dysfunction, endoplasmic reticulum stress, or dysregulation of calcium homeostasis [6]. In addition, non-cell autonomous processes such as neuroinflammation and cell-to-cell propagation of α-syn aggregation are involved in this disease [7,8]. These pathogenic mechanisms possibly act synergistically through complex interactions to promote neurodegeneration, yet a more complete elucidation of the exact molecular mechanisms acting in PD is required. In particular, determination of the transcription factors involved in these pathogenic pathways and the downstream gene effectors they induce will provide invaluable insights into a possible neuroprotective therapy.

CCAAT/enhancer binding protein β (C/EBPβ) is a member of the C/EBP family, which is part of the basic leucine zipper class of transcription factors. C/EBPβ mRNA has been observed in different brain regions of adult rodents [9], and it plays important roles in synaptic plasticity and memory formation, mainly in the hippocampus [10,11,12,13], neuronal differentiation [14,15], and hippocampal neurogenesis [16]. C/EBPβ also regulates the expression of several genes implicated in inflammatory response and injury [17,18,19,20,21,22,23]. Interestingly, mice lacking this transcription factor showed a decrease in inflammatory response and a reduction in pyramidal cell loss in the hippocampus after kainic acid injection [24,25]. Given the important roles of C/EBPβ in the brain, it is not difficult to envision an involvement of this transcription factor in neurodegenerative disorders. For instance, mRNA and protein levels of C/EBPβ are increased in AD patients’ brains [26,27] and in amyotrophic lateral sclerosis (ALS) spinal cord samples [28,29]; however, little is known regarding C/EBPβ implication on PD pathology. Our group has recently demonstrated that C/EBPβ reduction in in vitro and in animal models of PD results in an amelioration of dopaminergic damage and glial activation [30], suggesting that C/EBPβ depletion could constitute a valuable new therapeutic target for PD.

Mitochondrial transcription factor A (TFAM) is a nuclear encoded protein that is imported to mitochondria and is necessary for maintaining mitochondrial DNA (mtDNA) integrity and function as well as for cell survival [31,32,33]. Based on the way it binds to mtDNA, TFAM performs two main functions: when it binds specifically to promoter regions in mitochondrial mtDNA, it regulates transcription; when it binds nonspecifically around the entire mitochondrial genome, it has a structural function, compacting mtDNA into nucleoids [34]. A decrease of mtDNA and TFAM levels in patients and animal models of neurodegenerative disorders such as PD, AD, Huntington’s disease (HD), and ALS has been described (Reviewed in [35]), revealing the involvement of TFAM in neurodegeneration. Furthermore, a possible role of TFAM in neuroinflammation in the CNS has been observed, acting as a damage-associated molecular pattern (DAMP) outside the mitochondria [36,37].

Given that C/EBPβ is involved in the regulation of multiple pathways that are altered during neurodegenerative disorders, and more concretely, given that it is involved in neuroinflammation in cellular and animal models of PD, our objective was to further analyze the mechanisms regulated by this transcription factor in PD pathology. As a first step, we identified candidate genes possibly regulated by C/EBPβ in the context of PD by the combination of previous results from microarray studies in C/EBPβ-knock out (C/EBPβ−/−) mice brains and different in silico analyses. We then confirmed differential expression of several mitochondrial genes in the hippocampus of C/EBPβ−/− mice, TFAM being among them. Furthermore, we studied the effects of C/EBPβ overexpression and silencing on TFAM levels and mitochondrial activity in PD cellular models. Our results also demonstrate an accumulation of mitochondria in C/EBPβ depleted cells that may be due to the alteration of the autophagy pathway in the absence of this transcription factor.

## 2. Results

### 2.1. C/EBPβ Is Involved in the Regulation of Mitochondrial Genes in the Context of Parkinson’s Disease

Previous results from our laboratory and other groups have shown the important role of C/EBPβ in different processes in the nervous system, some of them being associated to certain pathologies (Reviewed in [38]). For this reason, we first wanted to deeply study which genes involved in neurodegenerative disorders, specifically in PD, could be regulated by C/EBPβ. In previous studies, we performed cDNA microarray analysis using hippocampal RNA isolated from C/EBPβ+/+ and C/EBPβ−/− mice and observed transcriptome modifications [19]. Based on these findings, we further studied published data on GWAS meta-analysis and RNA-seq in PD patients compared to controls [39,40,41,42,43]. We then searched for putative C/EBPβ binding sites in the −1300/+100 bp region from the transcription start site of selected genes by the use of UCSC genome browser [44], in order to find candidate genes involved in PD that could be regulated by this transcription factor. These bibliographic and in silico studies allowed us to generate a list of candidate genes implicated in multiple pathways such as mitochondrial function, inflammation, synapsis or vesicular transport and autophagy (Figure 1a).

Given the central role of mitochondria in neurodegenerative disorders, including PD [45], we focused on mitochondrial candidate genes for further studies. By RT-qPCR analysis of hippocampal samples from C/EBPβ+/+ and C/EBPβ−/− mice, we observed that the RNA levels of three mitochondrial genes were downregulated by 50% in mice lacking C/EBPβ (KO) compared to wild type (WT). These genes were *heme-binding protein 2* (*Hebp2*), *mitoferrin 1* (*Slc25a37*), and *mitochondrial transcription factor A* (*Tfam*) (Figure 1b and Appendix A).

### 2.2. C/EBPβ Regulates the Activity of Human TFAM Promoter

Given the implication of TFAM in neurodegenerative diseases such as PD [35], we next analyzed the regulation of this gene by C/EBPβ. In silico studies using ENCODE-UCSC, MatInspector, and MatchTM programs unraveled six putative C/EBPβ binding sites in TFAM promoter positions −911/−897, −885/−871, −661/−646, −594/−580, −424/−410, and −226/−97 (cut off value: 0.95), suggesting that C/EBPβ may directly regulate TFAM expression (Figure 1c). We next performed transient co-transfection experiments of a TFAM promoter-luciferase construct together with the pcDNA3-C/EBPβ expression plasmid in SH-SY5Y cells and, as shown in Figure 1d, the overexpression of C/EBPβ displayed a significant increase in the TFAM promoter activity. In order to localize the site by which this effect is generated, we next analyzed the response to C/EBPβ overexpression of different deletions of the TFAM promoter. Deletion of the sequence in position −490/−338 resulted in a significant reduction of C/EBPβ-induced luciferase activity (Figure 1d), possibly due to the elimination of the C/EBPβ binding site number 5 present in position −424/−410. As a conclusion, C/EBPβ activates the TFAM promoter activity either by directly binding to the sequence between −490/−338 positions or by activating a secondary transcription factor that binds to this region.

### 2.3. Possible Link between C/EBPβ and TFAM Protein Levels in Neuronal Differentiation and Oxidative Stress

Once we detected that C/EBPβ regulates the activity of the TFAM promoter, the next objective was to analyze the expression and localization of both proteins in neurons in physiological conditions as well as under neuronal differentiation and oxidative stress; two processes in which the implication of C/EBPβ has been described [15,30] and that are involved in PD pathology development [5]. First, we analyzed the effect of C/EBPβ overexpression in TFAM levels in the mouse N2A neuroblastoma cell line and in the previously generated p4 pool stably transfected with a pcDNA3-C/EBPβ expression plasmid [15]. These cells were grown in a normal and serum-free medium with the aim of studying the effect of C/EBPβ overexpression in the context of neuronal differentiation, as the absence of serum in media is an inductor of this process in neuroblastoma cells [46]. We first confirmed the overexpression of C/EBPβ in p4 cells by immunoblot, in which we observed a band of 45 kDa (corresponding to human C/EBPβ protein) that was not present in N2A cells. Interestingly, we also detected a band of 35 kDa in immunoblots with the anti-C/EBPβ antibody that was also present in N2A cells. This band probably corresponds to mouse endogenous protein in N2A cells and only in the case of p4 cells to post-translational variants of overexpressed human C/EBPβ protein (Figure 2a). C/EBPβ overexpression was corroborated by immunocytochemistry assays showing a major presence of C/EBPβ protein in the nuclei of p4 cells (Figure 2b). No changes were observed in TFAM levels in the absence of serum in N2A cells; however, C/EBPβ overexpression produced a significant increase of TFAM levels after 18 h of serum withdrawal (Figure 2a,b). These results suggest the participation of C/EBPβ in the regulation of TFAM levels when serum is absent from the medium.

Second, we studied the relationship between C/EBPβ and TFAM protein levels in a well-known PD cellular model consisting of SH-SY5Y dopaminergic cells treated with the neurotoxin 6-hydroxydopamine (6OHDA), which simulates the oxidative stress present in neurons during the disease [30,47,48,49]. C/EBPβ and TFAM protein levels along time were evaluated by immunoblot and immunofluorescence analyses in SH-SY5Y cells exposed to 6OHDA for 4, 6, 8, 18, and 24 h. C/EBPβ protein levels were low at basal conditions, but increased with time, reaching their maximum at 18 h (Figure 2c,d), when cells began to die due to stress. On the other hand, TFAM expression suffered a significant decrease after 4 h of treatment with 6OHDA, followed by a partial recovery of basal levels (Figure 2c,d). Since C/EBPβ activates the TFAM promoter (Figure 1d), it is plausible that this transcription factor may be involved in this recovery of TFAM levels. Given that the TFAM function is developed in mitochondria, we next analyzed whether the variation on its levels was accompanied with alterations in other mitochondrial proteins. We, hence, analyzed NADH:Ubiquinone oxidoreductase subunit B8 (NDUFB8) levels, a protein that is part of the respiratory chain complex I [50]. Interestingly, we found that NDUFB8 protein expression patterns resembled those of TFAM (Figure 2c).

### 2.4. C/EBPβ Silencing Produces TFAM and Other Mitochondrial Markers Accumulation in Cellular Models of Parkinson’s Disease and Neuroinflammation

To further determine how C/EBPβ regulates TFAM expression, we studied TFAM protein level variation in the absence of this transcription factor. As shown in Figure 3a,b, lentiviral transduction of SH-SY5Y cells with a shRNA sequence against C/EBPβ (sh-C/EBPβ) showed no significant differences in TFAM levels compared to control cells transduced with a non-targeting shRNA sequence as control (sh-Nt). These results were unexpected considering the fact that C/EBPβ activates the TFAM promoter; therefore, the lack of C/EBPβ would result in a decrease on TFAM protein levels. A possible explanation for this observation is that C/EBPβ silencing may be promoting the accumulation of mitochondria in cells. Then, even if TFAM was downregulated in silenced cells, it would be accumulated in these organelles. To test this hypothesis, we studied, again, the levels of the mitochondrial protein NDUFB8, and we observed a significant increase in sh-C/EBPβ cells when compared with sh-Nt cells in the presence of 6OHDA for 8 and 18 h (Figure 3a). Moreover, immunofluorescence analysis showed a significant increase in the levels of TOM20, a component of the mitochondrial outer membrane (Figure 3b). Altogether, these results indicate an accumulation of mitochondria in the absence of C/EBPβ in dopaminergic neurons under oxidative stress.

The development and progression of PD are affected by cells other than neurons (glia), as well as stress conditions other than oxidative stress. Therefore, we next studied the effect of C/EBPβ silencing on TFAM protein levels in astroglial primary cultures. We also analyzed whether inflammation induction by the treatment of cells with bacterial lipopolysaccharides (LPS, 10 µg/mL) had any effect on TFAM levels, due to their role as an inductor of C/EBPβ expression [19,51]. As shown in Figure 3c,d, lentiviral mediated silencing of C/EBPβ in mouse-derived astrocytes (sh-C/EBPβ) resulted in an increase of TFAM levels that reached significance after LPS treatment when compared to control astrocytes transduced with shRNA against luciferase (sh-Ctl). This increase was also observed when other mitochondrial proteins such as ATPB (a subunit of respiratory chain complex V) (Figure 3c) and TOM20 (Figure 3d) were studied, again suggesting an accumulation of these organelles.

### 2.5. Mitochondria Accumulation does Not Result in an Increase on Mitochondrial Function

Our next aim was to elucidate whether these mitochondria accumulated in the absence of C/EBPβ in neurons and astrocytes had a normal activity. To answer this question, we measured mitochondrial membrane potential (ΔΨm) that reflects mitochondrial health and function. By the use of tetramethyl-rhodamine methylester (TMRM) as a fluorescent indicator of ΔΨm, we observed a significant decrease of this parameter in SH-SY5Y cells under oxidative conditions (Figure 4a). Furthermore, ΔΨm was lower in the absence of C/EBPβ and oxidative stress did not sum to this effect (Figure 4a). These results suggest that C/EBPβ silencing results in an alteration of mitochondrial function in dopaminergic neurons that is not recovered by the previously observed accumulation of these organelles in silenced cells under oxidative stress (Figure 3a,b).

No differences were detected when studying ΔΨm in C/EBPβ-downregulated astrocytes and/or under inflammatory stress conditions (Figure 4b), then suggesting that C/EBPβ may not be necessary for normal mitochondrial function in astrocytes.

According to these results, mitochondria accumulation in silenced cells (Figure 3) does not lead to an increase in full-functioning mitochondria in astrocytes and SH-SY5Y cells, suggesting that the mitochondria accumulated may not be functional.

### 2.6. C/EBPβ Depletion Causes the Alteration in Autophagy in Cellular Models of Parkinson’s Disease and Neuroinflammation

The decrease in C/EBPβ levels results in the accumulation of different mitochondrial markers under stress conditions but does not increase the presence of active mitochondria in cells. One possible explanation is that in C/EBPβ silenced cells, damaged or inactive mitochondria are not properly degraded and accumulate inside the cell. One of the principal mechanisms for damaged mitochondria renewal is mitophagy, the selective autophagic degradation of mitochondria, which is needed for proper mitochondrial homeostasis control [52]. Macroautophagy (hereafter referred to as autophagy) is a mechanism responsible for the clearance and recycling of damaged or redundant long-lived proteins and organelles by engulfing cytoplasmic materials into a double membrane vesicle and targeting them to lysosomes for degradation [53]. As showed in Figure 1a, some of the candidate genes to be regulated by C/EBPβ are involved in autophagy, which made us infer that the decrease of this transcription factor levels may result in the alteration of this mechanism. If autophagy does not work properly, mitochondria degradation does not occur in a normal way, and this could be causing the observed accumulation of these organelles. To test this hypothesis, we studied whether the reduction of C/EBPβ levels altered autophagy in the different cellular models of PD and neuroinflammation. Immunoblot and immunofluorescence analyses were performed to study the levels and distribution of two of the most commonly used autophagy markers; microtubule-associated protein light chain 3 (LC3) and sequestrosome 1 (p62 or SQSTM1). The LC3 cytosolic isoform (LC3-I) is conjugated to phosphatidyletanolamine in phagophore and autophagosomal membranes and becomes a lipidated form called LC3-II, meaning that LC3-II levels are proportional to the amount of autophagic vesicles (AVs) [54]. On the other hand, p62/SQSTM1 is a ubiquitin-binding protein that is incorporated to AVs by its direct interaction with LC3, and it is degraded by the autophagy process [55]. Then, an accumulation of p62 indicates an alteration in autophagic activity. As shown in Figure 5a, there is an increasing trend of LC3-II levels in C/EBPβ silenced cells when compared to sh-Nt, which is higher under oxidative stress conditions, reaching significance after 18 h of 6OHDA treatment. Similar results were observed when p62 levels were measured; however, this protein was significantly accumulated in control cells after 18 h of 6OHDA treatment. Given that p62/SQSTM1 levels can be transcriptionally regulated by different cellular processes such as oxidative stress [56], the accumulation of this protein in cells may not be exclusively dependent on the autophagy process. For this reason, we studied this protein distribution in cells by immunofluorescence. In this way, we observed two different p62/SQSTM1 fluorescence distribution patterns; a diffuse cytoplasmic pattern, and another one with the appearance of punctiform structures that primarily represent AVs (Figure 5b). The first pattern was more abundant in silenced cells and/or in 6OHDA treated cultures; therefore, we hypothesized that this may be due to an increase in p62 expression associated to oxidative stress or to a higher presence of misfolded proteins or small protein aggregates that are not included in AVs. Once we defined these different fluorescence patterns, we measured total p62/SQSTM1 fluorescence intensity (Figure 5c) and the number of round particles (Figure 5d) per cell. Thus, similar results to the immunoblot analysis were observed when we measured total fluorescence intensity. Moreover, the number of p62/SQSTM1-containing particles was significantly higher in silenced cells when compared with sh-Nt in basal conditions and after the treatment with 6OHDA for 8 and 18 h. Altogether, these results show that 6OHDA treatment causes the accumulation of different autophagy markers in SH-SY5Y cells and that this accumulation is higher in C/EBPβ silenced cells. Given that the accumulation of mitochondrial markers was also observed in astrocytes, we followed the same reasoning as for SH-SY5Y cells and we studied whether autophagy was also altered in these glial cells. Figure 5e shows a significant accumulation of LC3-II in C/EBPβ silenced cells that was higher after inflammation induction. Regarding p62/SQSTM, LPS treatment caused an increase in its levels, which tend to be higher in C/EBPβ silenced cells. Again, the lack of a significant difference in p62/SQSTM levels between sh-Ctl and sh-C/EBPβ cells in an inflammatory context may be due to the transcriptional regulation of the gene that codifies this protein in stress conditions [56]. However, in this cell type, the analysis of the number of p62/SQSTM1-containing particles per cell for a better estimation of AVs accumulation by p62/SQSTM fluorescent staining was not possible due to the clear difference in size between cells.

Importantly, as autophagy is the net result of autophagosomes formation and degradation, this rise in autophagic markers levels can be due either to an increase in new autophagosome formation, that is, an increase in autophagy induction; or to an alteration in the autophagy pathway that avoids autophagosomes degradation (Figure 6a). For this reason, a more accurate way to study whether there is an alteration in autophagic activity in our experimental model is to measure the autophagic flux, which reflects the total amount of AVs that are delivered to and degraded by the lysosomes. Autophagic flux can be measured as the difference of autophagy marker levels in the presence and absence of lysosomal inhibitors [57]. In this work, we used the lysosomal inhibitor chloroquine (CQN, 30 µM), a compound that is trapped in acidic environments, as lysosomes, causing an increase in their pH and, therefore, inhibiting the activity of the degradative enzymes [58]. Immunoblot analysis of LC3-II showed that CQN caused a marked increase in its levels under all conditions tested (Figure 6b). We next calculated the autophagic flux in the different experimental groups by subtracting the value of LC3-II in the absence of CQN to the value of this protein level in the presence of the inhibitor. As shown in Figure 6c, no significant differences in autophagic flux were detected between the experimental groups, although a trend of an increase of autophagic flux with 8 h of 6OHDA treatment was observed in sh-Nt and sh-CEBPβ cells. This trend is probably due to the high variability occurring in this experimental approach that is mainly due to the high velocity of the autophagy process in the SH-SY5Y cell line, which results in very low LC3-II protein levels in basal conditions [59]. This fact makes very difficult its detection by immunoblot, generating a high variability between experiments depending on the capacity of measuring LC3-II basal levels in each occasion. For this reason, we also measured autophagic flux with p62/SQSTM marker and again no significant differences were observed (Appendix A). In this case, a decreasing trend in autophagy flux is observed in sh-Nt cells after 6-OHDA treatment during 18 h, which does not occur in sh-C/EBPβ cells. A possible explanation for this is that, as abovementioned, oxidative stress may be transcriptionally regulating p62/SQSTM1 levels [56]. The lack of significant differences in autophagic flux indicates that the AVs accumulation observed in sh-C/EBPβ cells under oxidative stress conditions is not due to an increase in autophagy induction but to an alteration in their degradation. Then, C/EBPβ could participate in the regulation of genes involved in the AVs degradation step of the autophagy process, which are induced by 6OHDA in SH-SY5Y cells. In a similar manner, CQN produced an increase in LC3-II levels in astroglial cultures in all studied conditions (Figure 6c). Similar to that performed in SH-SY5Y cells, we calculated the autophagic flux with no significant differences between the studied experimental groups (Figure 6e). This indicates that, also in astrocytes, the accumulation of AVs previously described is due to a failure in their degradation but not to an increased induction of their formation. However, it is important to consider that in silenced astroglial cultures, there is a trend of decrease in autophagic flux, which may be due to a failure in the autophagy induction in this cell type.

## 3. Discussion

The role of C/EBPβ in different neuronal injury processes such as neuroinflammation or excitotoxicity has been widely described. Furthermore, the implication of this transcription factor in different neurodegenerative disorders as AD [26,27], HD [60], or ALS [28,29] has also been reported. Regarding PD, the mitigation of glial activation and neurodegeneration in an animal model of the disease by C/EBPβ knock down has been recently described by our group [30], suggesting that this transcription factor could be a good therapeutic target for this disease. The knowledge about the role of C/EBPβ and its downstream genes in PD is currently increasing. For instance, it has been described that C/EBPβ is involved in the regulation of the genes codifying α-synuclein and dopamine metabolism enzyme Monoamine oxidase B (MAO-B) [61,62,63]. Moreover, C/EBPβ overexpression facilitates PD pathologies and elicits motor disorders and constipation in mice treated with toxins that alter the mitochondrial electron transport chain (ETC) [64] and in WT and α-synuclein transgenic mice. [63,65]. Interestingly, C/EBPβ-binding sites have been detected surrounding genome locations that codify some miRNAs that are dysregulated in PD patients [66,67]. In this work, we performed bibliographic and in silico studies that, combined with previous microarray results obtained from C/EBPβ−/− mice brains, rendered a list of candidate genes to be regulated by C/EBPβ in the context of PD. We focused on mitochondrial genes for further studies because of the implication of these organelles in this disease pathology through mechanisms such as calcium homeostasis alteration or oxidative stress induction [45]. RT-qPCR analysis performed in C/EBPβ−/− mice hippocampus samples confirmed the downregulation of three mitochondrial genes; *Tfam*, *Hebp2*, which is involved in the mitochondrial permeabilization for necrotic cell death induction [68], and *Slc25a37*, an iron importer that has a role in heme and iron-sulfur clusters synthesis in mitochondria fundamentally in high heme synthesis requiring conditions [69,70]. Accordingly, the study of ΔΨm by TMRM showed that the absence of C/EBPβ in neurons resulted in the alteration of mitochondrial activity in basal conditions.

Nowadays, the involvement of C/EBPβ in mitochondrial activity is not clear. While some authors have described that the absence or decrease of this transcription factor is accompanied by an increase in mitochondrial genes expression in white adipose tissue [71], liver [72], or hypertrophic myocardium [73] normally in pro-inflammatory contexts; other authors have observed that C/EBPβ increases mitochondrial biogenesis or improves mitochondrial activity in white adipose tissue [74], in kidney tubular epithelial cells [75], and in human lung fibroblasts [76]. Regarding the nervous system, C/EBPβ overexpression has been shown to improve mitochondrial activity through IGF-1 transcription increase in dorsal root ganglia in the diabetes context [77]. However, in PD, the mitochondria ETC failure induces the expression of this transcription factor, although it does not result in a mitochondrial activity recovery due to the activation of other harmful pathways in dopaminergic neurons [65]. To our knowledge, in this work, we describe for the first time the implication of C/EBPβ in the basal mitochondrial function in dopaminergic neurons. These data together show the complex regulation exerted by C/EBPβ, which could be contradictory in a particular process depending on the cell type or cellular context.

Once we detected that C/EBPβ was involved in some mitochondrial genes’ regulation, we further studied the effect of the overexpression of this transcription factor on *TFAM* gene promoter activity. Although *HEBP2* and *SLC25A37* have been described to be involved in different diseases such as cancer [78,79,80,81,82,83], anemia [84,85], porphyria [70,86], or depression [87], we focused on *TFAM* for further studies due to its widely observed implication in different neurodegenerative disorders, PD being among them. We then observed that C/EBPβ induced the activity of the *TFAM* promoter in the −497/−338 region. Although there is a C/EBPβ binding site in this DNA sequence, it has not been demonstrated in this work whether the increase in TFAM promoter activity by overexpression of C/EBPβ is due to its direct interaction with this region or to the activation of a secondary transcription factor acting in this region; both options are possible. In line with our results, a direct regulation of TFAM transcription by C/EBPβ in the inflammatory context induced in acute kidney injury has also been described [88].

Given that C/EBPβ is involved in the TFAM promoter activity regulation, we next studied the variation of TFAM protein levels in the presence and absence of C/EBPβ in different neuronal cell lines and astroglia primary cultures. Despite the fact that there was an increase on TFAM protein levels in neurons overexpressing C/EBPβ in serum withdrawal conditions, the absence of this transcription factor also resulted in TFAM accumulation in SH-SY5Y as well as in astrocytes, fundamentally in oxidative stress and neuroinflammatory conditions, respectively. These results could be confusing, considering that if C/EBPβ increases *TFAM* promoter activity the absence of this transcription factor would be expected to produce a decrease on TFAM protein levels. A possible explanation is that the knock-down of C/EBPβ may be causing the accumulation of mitochondria leading to the accumulation of TFAM inside these organelles. The study of the levels of other constitutive mitochondrial proteins (NDUFB8, TOM20, and ATPB) confirmed this hypothesis, as they were also accumulated when C/EBPβ levels were lower in the different studied stress conditions, as observed with TFAM. Moreover, the analysis of ΔΨm showed that these mitochondria accumulations did not result in an increase of mitochondrial activity in C/EBPβ silenced cells. It is probable that the way in which C/EBPβ absence causes these organelles accumulations is by the alteration of their biogenesis-degradation balance, which could result in the accumulation of non-functional mitochondria (Figure 7a). Since autophagy is one of the main processes for damaged mitochondria degradation, our next aim was to determine whether this process was altered in C/EBPβ knock-down cultures. In such conditions, we observed an accumulation of autophagic markers in SH-SY5Y and astroglial cells under oxidative stress and in an inflammatory context, respectively. Furthermore, it was also observed that the autophagic flux was not different between silenced and non-silenced cells, indicating that the accumulation of autophagic markers was not due to an increase in the autophagy pathway induction, but to a decrease in autophagosome degradation. Then, we can conclude that C/EBPβ is involved in the regulation of autophagy process in cellular models of PD, which may explain the accumulation of TFAM and other mitochondrial markers in C/EBPβ silenced cells. In accordance with these results, some authors have described the participation of C/EBPβ in the regulation of genes involved in autophagy in hepatocytes [89], adipocytes [90,91,92], and zebrafish [93], and also during parasite elimination process in immune [94,95] and non-hematopoietic cells [96]. Related to our results, Yang et al. showed that C/EBPβ is involved in the protective role of Sestrin 2 (SESN2) upon muscular atrophy caused by denervation, which, among other effects, improves mitochondrial activity via mitophagy activation [97]. However, the implication of C/EBPβ in autophagic process in the CNS is still little known. In this regard, Gómez-Santos et al. [61] described that the increase of this transcription factor level induced by dopamine in SH-SY5Y cells was accompanied by an increase in the number of Avs; however, they did not show a direct relationship between these two events. More recent studies in neurons treated with methamphetamine indicated that C/EBPβ silencing decreased the autophagy induced by this drug [98,99].

Finally, it is important to note that the present work and other studies show that C/EBPβ is an important transcription factor for the correct functioning of neurons, as it is involved in neurogenesis, differentiation, mitochondrial function, and autophagy in these cells [15,16,100,101]. However, our group has previously demonstrated that in an in vivo PD model, C/EBPβ downregulation contributes to neuronal survival [30]. These seemingly contradictory effects may be due to the fact that, in the existing inflammatory context in the animal model of primarily glia-induced PD, the role of C/EBPβ as a proinflammatory factor is stronger than its involvement in neuronal physiological functions. Therefore, C/EBPβ silencing would be more beneficial than detrimental for neuron survival in these PD-like conditions. According to this, some authors have observed either that C/EBPβ downregulation reduces neuronal death promoted by different stressors such as growth factor withdrawal or N-methyl-D-aspartic acid (NMDA) [102], and that this transcription factor has a neuroprotective role against different neuronal death inductors [71,103].

As a conclusion, C/EBPβ could be a good therapeutic target for the treatment of PD. However, to avoid possible side effects derived from this transcription factor’s physiological functions in neurons, one suitable approach would be the silencing of C/EBPβ by the specific interference RNA (RNAi) delivery in astroglia and microglia, which are strongly involved in the inflammatory process occurring in PD (Figure 7b). Interestingly, science is advancing in the field of targeted therapy by the use of different techniques such as adeno associated viruses [104] or exosomes [105]. Moreover, studying the genes regulated by C/EBPβ that are involved in different harmful molecular pathways occurring in PD would give us more clues about this transcription factor mechanism of action and about new proteins that could be targeted for the treatment of Parkinsonism.

## 4. Materials and Methods

### 4.1. Animals

Hippocampal RNA was extracted from adult C/EBPβ+/+ and C/EBPβ−/− mice, which were generated from heterozygous breeding pairs, kindly provided by C. M. Croniger and R. W. Hanson (Case Western Reserve University) [106]. Genotypes were identified by PCR in genomic DNA extracted from mice tails by the use of the REDExtract-N-AmpTM tissue PCR kit (Sigma-Aldrich, Madrid, Spain). Primary astroglial cultures were isolated from neonatal C57BL/6J mice (2 days old), obtained from our animal facilities in the *Instituto de Investigaciones Biomédicas* Alberto Sols. All animals were bred with food and water *ad libitum* in a 12 h dark–light cycle. Experimental procedures were approved by the Ethics Committee for Animal Experimentation of the *Consejo Superior de Investigaciones Científicas* (CSIC) (protocol code PROEX135/18, approved on 21/1/2019) and carried out in accordance with the European Communities Council (directive 2010/63/UE) and National regulations (RD.1386/2018). Experiments with animals were designed following the Three Rs Principle (replacement, reduction, and refinement) and special care was taken to avoid animal suffering.

### 4.2. Real-Time Quantitative PCR

Total RNA samples were extracted from hippocampus of C/EBPβ+/+ and C/EBPβ−/− using TRIzol (Agilent Technologies, Santa Clara, CA, USA). RNA samples (2µg) were used for cDNA synthesis by reverse transcription with random primers pd(N)6 (Promega, Madrid, Spain), 1 mM dNTPs, and 160 U of the reverse transcriptase M-MLV (Promega) in the presence of 20 U of RNAse inhibitor (Promega). Gene expression variations were confirmed by RT-qPCR using Syber Green (Applied Biosystems, Madrid, Spain) and 800 nM concentrations of the specific primers to detect HEBP2, SLC25A37, or TFAM mRNAs (listed in Table 1), in Step One Plus thermal cycler (Applied Biosystems). Relative mRNA levels were measured in triplicate for each gene in all samples by the use of 2^−ΔΔCt^ method [107] and the housekeeping 18S ribosomal RNA was used for normalization.

### 4.3. Cell Cultures

#### 4.3.1. Cell Lines

Wild type and C/EBPβ overexpressing neuroblastoma Neuro2A (N2A) mouse cells previously generated [15] were grown in DMEM media supplemented with 10% fetal bovine serum (FBS), 2 mM L-Glutamine, 100 U/mL penicillin, and 100 µg/mL streptomycin. Cells containing C/EBPβ overexpressing plasmid were selected with 0.5 mg/mL geneticin (G418, Gibco, Madrid, Spain). Human neuroblastoma SH-SY5Y cells were purchased from ATCC (Ref.CRL-2266) and were propagated and maintained in RPMI media supplemented with 10% FBS, 2 mM L-Glutamine, and 100 µg/mL gentamicin (Genta-Gobens, Normon, Madrid, Spain). All cells were cultured in a cell incubator (ShelLab, Cornelius, OR, USA) at 37 °C, 90% humidity, and 5% CO_2_. As a model of dopaminergic cytotoxicity, cells plated into 60-mm or 24-wells plates were treated with 6-hydroxydopamine (6OHDA, 35 μM, Sigma) during 4, 6, 8, 18, or 24 h.

#### 4.3.2. Primary Astroglial Cultures

Primary glial cell cultures were obtained from 2-day-old C57BL6/J mice cerebral cortex as previously described [108]. Briefly, animal brains were extracted and dissected to isolate the cerebral cortex, which was mechanically dissociated and centrifuged at 1050 rpm. The cellular pellet was plated in poly-D-lysine (20 µg/mL) pretreated flasks (75 cm^2^) with DMEM supplemented with 10% FBS, 10% horse serum, 100 U/mL penicillin, and 100 µg/mL streptomycin. These mixed glial cultures were grown for 10 days in standard conditions (37 °C, 90% humidity, and 5% CO2). Different cellular types were later separated by agitation of cell cultures in an orbital shaker for 4 h at 230 rpm and 37 °C, to isolate non-adherent microglial cells in the media, followed by an overnight agitation at 260 rpm and 37 °C, to isolate oligodendrocytes and some remaining microglial cells. Finally, only astrocytes adhered to flasks surface were collected and plated onto 100-mm plates with 1:1 DMEM/HAM’S-F12 media, supplemented with 10% FBS, 2 mM glutamine and 100 µg/mL gentamicin. The purity of the cultures was >95%, as determined by immunofluorescence analysis using an antiglial fibrillary acidic protein (GFAP; clone G-A-5; Sigma-Aldrich) antibody to identify astrocytes, an anti-Iba1 (Wako, Fujifilm Wako Chemicals Europe, Neuss, Germany) antibody to identify microglial cells, and an anti-O4 (Millipore, Madrid, Spain) antibody as an oligodendrocyte marker as previously described [49]. Only astroglial cultures were used in this study. Therefore, primary astrocytes were plated into 60-mm or 24-wells plates and treated with LPS (10 μg/mL) during 24 h, to induce inflammation.

### 4.4. TFAM Promoter Cloning and Luciferase Assay

Different fragments of the human TFAM promoter were PCR-amplified from human genomic DNA using specific primers (listed in Table 2) and Platinum SuperFi polymerase (Invitrogen, Madrid, Spain). These fragments corresponded to the long promoter (pTFAM/1188) and different deletions eliminating putative C/EBPβ binding sites detected in silico by the use of ENCODE-UCSC and MatchTM databases (pTFAM/652, pTFAM/500, pTFAM/352, and pTFAM/270). These sequences were directly cloned in pCR-Blunt vector (Invitrogen) and, after sequencing, they were subcloned in the promoterless luciferase reporter vector PGL4.10. For transient transfection experiments, semiconfluent SH-SY5Y cells were grown in 24-wells plates and transfected with 2 µL/mL ScreenfectA-Plus (Screenfect, Eggenstein-Leopoldshafen, Germany) using the above-described constructs (0.5 µg DNA/well) in the presence or absence of a C/EBPβ overexpression plasmid pcDNA3-C/EBPβ (1 µg DNA/well). Luciferase activity was measured 24 h post-transfection by the use of Luciferase Assay System kit (Promega) and β-galactosidase activity was used to determine transfection efficiency. The transient transfection experiments were repeated at least three times in triplicate.

### 4.5. Lentiviral-Mediated Gene Silencing

To knockdown C/EBPβ expression, SH-SY5Y cells and mouse primary astrocytes were infected with lentiviral particles expressing a shRNA specific for human or mouse C/EBPβ, respectively. For SH-SY5Y cells, we used a commercial plasmid (PLKO.1) containing a shRNA sequence against C/EBPβ (sh-C/EBPβ), 5′-CCGGCGACTTCCTCTCCGACCTCTTCTCGAGAAGAGGTCGGAGAGGAAGTCGTTTTT-3′ (Sigma Mission^®^ ShRNA-Ref. TRCN0000007442). A non-targeting silencing sequence (Sigma Mission^®^ Control Vectors-Ref. SHC002) was used as a control (sh-Nt). The lentiviral particles were obtained in HEK293T cells that were transfected with the appropriate lentiviral expression vectors and the third-generation packaging vectors pMD2-G, pMDLg/pRRE, and pRSV-Rev [109]. The medium containing lentiviruses was recovered, filtered through a 0.45 µm filter, and added to the recipient cells. The same procedure was repeated 8 and 24 h later.

In the case of the astrocytes, the interfering sequence: 5′-GAGCGACGAGTACAAGATGTTCAAGAGACATCTTGTACTCGTCGCTCTT-3′ was cloned between BamHI/EcoRI sites into pGreenPuroTM shRNA Cloning and Expression Lentivector (System Biosciences, Palo Alto, CA, USA), according to the manufacturer’s protocol, in which shRNA was expressed under the control of the H1 promoter. The pGreenPuro™ construct with the luciferase shRNA template provided by System Biosciences was used as control in all the experiments (sh-Ctl). Lentivirus particles were again generated in HEK293T cells and concentrated by ultracentrifugation as previously described [110]. A multiplicity of infection of 3 particles/cell was used for astrocyte infection after viral tittering.

### 4.6. Immunoblot Analysis

Astrocytes, N2A and SH-SY5Y cells were plated into 60-mm dishes with a 90% confluence, and, after the corresponding treatments, they were harvested for protein extraction with RIPA buffer (25 mM Tris-HCl pH = 7.6, 150 mM NaCl, 1% NP-40, 1% sodium deoxycholate and 0.1% SDS) containing protease and phosphatase inhibitors. A total amount of 20–30 μg of protein was separated on a 12% or 15% SDS-PAGE and transferred into PVDF membranes (GE Healthcare, Hoevelaken, The Netherlands). The membranes were blocked in T-TBS (20 mM Tris-HCl pH = 7.5, 150 mM NaCl and 0.1% Tween-20) with 5% skimmed milk, incubated with primary and secondary antibodies, and washed according to standard procedures. The following primary antibodies were used: mouse monoclonal anti-C/EBPβ (C19 clone, Santa Cruz, CA. USA, ref. sc7962), mouse anti-TFAM (Santa Cruz, ref. sc166965), mouse anti-LC3 (Santa Cruz, ref. sc376404) mouse anti-ATPB (Abcam, ref. ab14730), rabbit anti-p62 (Sigma, ref. P00067), mouse anti-α-tubulin (Sigma, ref. T9026), and mouse anti-α-actin (Sigma, ref. A5441). Secondary peroxidase-conjugated goat anti-mouse and anti-rabbit antibodies (Jackson ImmunoResearch, Cambridgeshire, UK) were used. Luminescence was captured by the Fusion^®^ (Vilber, Marne-la-Vallée, France) device and quantified by the use of Image-J software [111]. The figures show representative images of at least three independent experiments. Values are expressed as the mean of the measured chemiluminescence signal.

### 4.7. Immunocytochemistry

The different cell types were seeded in 24-well plates containing poly-D-lysine (20 µg/mL) pretreated glass coverslips. After treatments, they were fixed for 20 min with 4% paraformaldehyde and blocked for 30 min at 37 °C with 5% goat serum in a solution of 1% Triton X-100 in PB. After that, cells were incubated for 1 h at 37 °C with different primary antibodies: mouse anti-C/EBPβ (C19 clone, Santa Cruz, ref. sc7962), rabbit anti-TFAM (Cell Signaling, Leiden, The Netherlands, ref. 8076), or rabbit anti-p62 (Sigma, ref. P00067). Cover-slips were then washed with PB and incubated for 45 min at 37 °C with Alexa-labeled secondary antibodies (Thermo Fisher, Madrid, Spain). Staining of nuclei was performed using 4′,6-Diamidino-2-phenylindole (DAPI). Images were acquired in a LSM710 laser scanning spectral confocal microscope (Zeiss) and the microscope settings were adjusted to produce the optimum signal-to-noise ratio. Quantitative analysis of fluorescence intensity and particles number was performed by the use of Image-J software, and at least five different counting fields were randomly selected from three independent experiments.

### 4.8. Mitochondrial Membrane Potential (ΔΨm) Measurement

Astrocytes and SH-SY5Y cells were grown on 25 mm coverslips to carry out the corresponding treatments the day of the ΔΨm acquisition, cells were incubated with 40 nM tetramethyl-rhodamine methyl ester (TMRM) in a HEPES-buffered salt solution (HBSS; 156 mM NaCl, 3 mM KCl, 2 mM MgSO_4_, 1.25 mM KH2PO_4_, 2 mM CaCl_2_, 10 mM glucose and 10 mM HEPES; pH = 7.35) for 40 min using the Attofluor Cell Chambers (Thermo Fisher). The dye was present in the chamber at the time of image acquisition. TMRM, a lipophilic fluorescent dye, is accumulated in active mitochondria and its fluorescence intensity is directly proportional to ΔΨm [112]. Therefore, a reduction in TMRM fluorescence represents mitochondrial depolarization. Confocal images were obtained using a LSM710 laser scanning spectral confocal microscope (Zeiss, Oberkochen, Germany) equipped with a META detection system and a ×40 water immersion objective. TMRM was excited using the 560 nm laser line and fluorescence was measured above 580 nm. The Z-stack images were analyzed using the Volocity Quantitation software 6.5.1 (Quorum Technologies, Puslinch, ON, CANADA) and TMRM values for control cases were set to 100%.

### 4.9. Statistical Analysis

GraphPad Prism 6.01 and SPSS version 26 (IBM corporation, Armonk, NY, USA) software were used for statistical analysis. Comparisons between the mean of two continuous variables with normal distribution were performed by *t*-test. Statistical significance for multiple comparisons among more than two experimental groups was performed by one-way ANOVA followed by Bonferroni correction, after checking the normality and homoscedasticity. Data are shown as the mean of at least three experiments ±standard error (SE) or ±standard deviation (SD) as indicated in the figure legends. Differences between the experimental groups were considered statistically significant when *p* < 0.05.

## Figures and Tables

**Figure 1 ijms-24-01459-f001:**
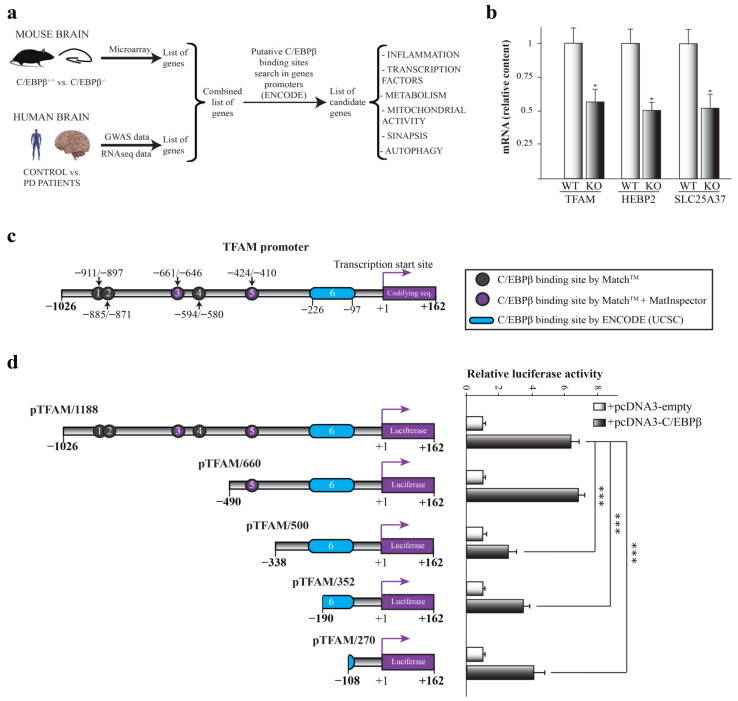
Implication of C/EBPβ in the regulation of mitochondrial genes in the context of Parkinson’s disease. (**a**) Scheme of the process followed for the generation of a list of candidate genes regulated by C/EBPβ related to PD. Previous microarray data obtained in C/EBPβ−/− mice were combined with published data of genome-wide association study (GWAS) and RNA-seq in PD patients vs. controls. Then, C/EBPβ binding sites were searched for among the promoters of genes of the resulting list by the use of the ENCODE-UCSC database. Several candidate genes involved in PD and possibly regulated by C/EBPβ were identified. (**b**) RT-qPCR analysis of C/EBPβ+/+ (WT) and C/EBPβ−/− (KO) mice hippocampus mRNA of mitochondrial genes *Tfam*, *Hebp2,* and *Slc25a37*. Data are shown as the mean ± SE of the RNA relative levels in n = 10 WT animals and n = 5 KO animals. * *p* ≤ 0.05 (t-student). (**c**) In silico determination of putative C/EBPβ binding sites on TFAM promoter by the use of MatchTM, MatInspector, and ENCODE (UCSC) program as indicated. (**d**) Effect of C/EBPβ overexpression on the activity of different TFAM promoter segments analyzed by transient transfection in SH-SY5Y cells. The data represent the mean ± SE of luciferase activity referred to basal and determined in triplicate in at least four independent experiments. *** *p* ≤ 0.001 (ANOVA-Bonferroni).

**Figure 2 ijms-24-01459-f002:**
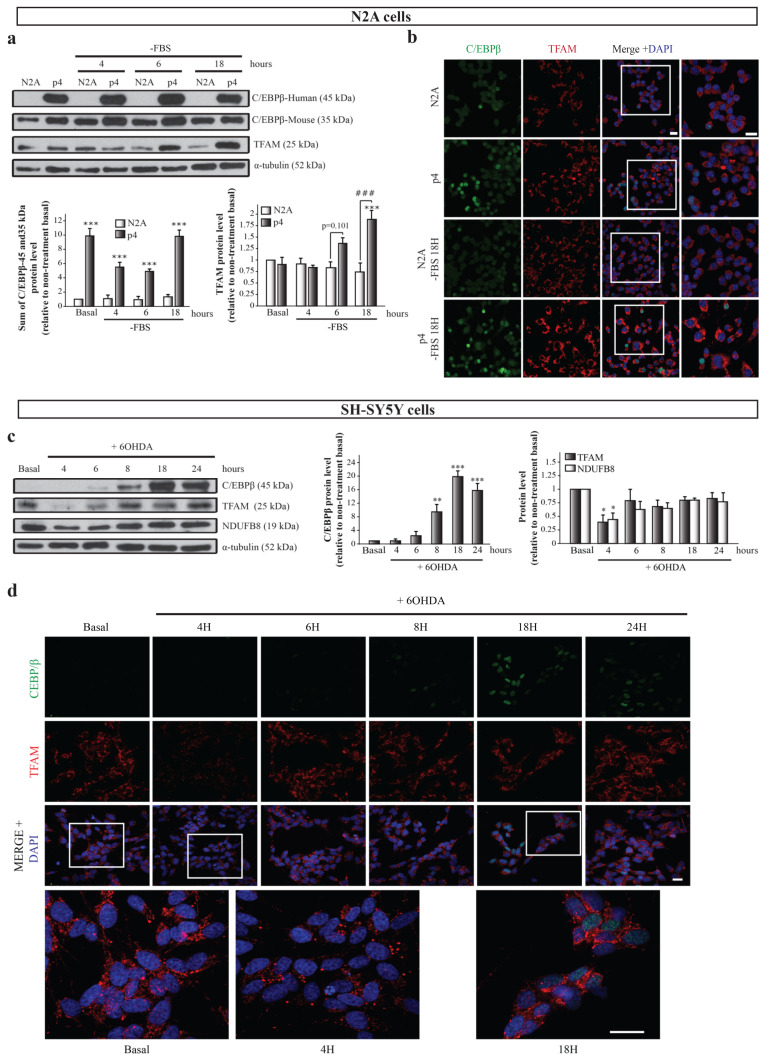
Relation of C/EBPβ and TFAM protein levels in neurons under differentiation or oxidative stress conditions. N2A and p4 (overexpressing C/EBPβ) cells were grown in serum free medium (-FBS) to induce differentiation for 4, 6, and 18 h. (**a**) Representative immunoblots and quantifications showing human (45 kDa) and mouse (35 kDa) C/EBPβ, TFAM, and α-tubulin protein levels in these cells. Data are presented as the mean of the relative protein levels normalized by α-tubulin levels in at least four independent experiments ± SE. *** *p* ≤ 0.001 vs. non-treatment basal, ### *p* ≤ 0.001 vs. control N2A cells with the same treatment (ANOVA-Bonferroni). (**b**) Representative immunofluorescence images showing C/EBPβ (green) and TFAM (red) expression in the same cell cultures. Nuclei were stained with DAPI (blue). Insets show high magnification of the selected area. Scale bar = 20 µm. (**c**) Dopaminergic cell line SH-SY5Y cultures were treated with 6-hydroxydopamine (6OHDA, 35 µM) for 4, 6, 8, 18, and 24 h to simulate oxidative stress present in PD and immunoblot analysis was performed to determine C/EBPβ, TFAM, and NDUFB8 protein levels. Data on the graphs correspond to the mean of the relative levels of these proteins normalized by α-tubulin in four independent experiments ± SE. * *p* ≤ 0.05, ** *p* ≤ 0.005, *** *p* ≤ 0.001 (ANOVA-Bonferroni). (**d**) Representative images of C/EBPβ (green) and TFAM (red) immunofluorescence analyses in the same cells are shown with DAPI stained nuclei. Insets show high magnification of the selected area. Scale bar = 20 µM.

**Figure 3 ijms-24-01459-f003:**
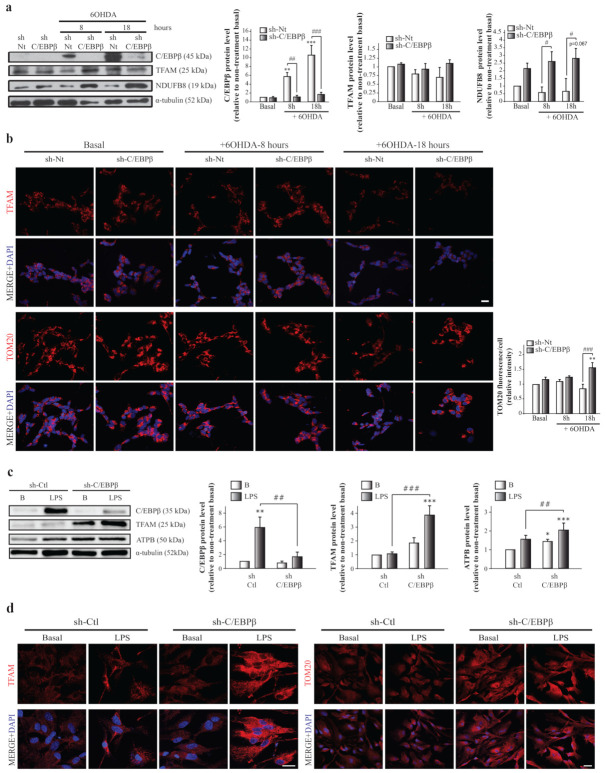
TFAM and mitochondrial protein accumulation in SH-SY5Y and astrocytes with reduced levels of C/EBPβ. Dopaminergic cell line SH-SY5Y cultures were infected with lentiviral particles containing a Non-Targeting shRNA (control, sh-Nt) or with shRNA targeting C/EBPβ (sh-C/EBPβ). Cell cultures were treated with 35 µM 6OHDA for 8 and 18 h and (**a**) immunoblot analysis were performed to determine C/EBPβ, TFAM, and NDUFB8 protein levels. Data on the graphs correspond to the mean of the relative levels of these proteins normalized by α-tubulin in four independent experiments ± SE. (**b**) Representative images of TFAM and TOM20 (red) immunofluorescence analyses in the same cells with DAPI stained nuclei. Scale bar = 20 µM. The bar graph shows the mean of the relative TOM20 fluorescence intensity per cell ± SE. ** *p* ≤ 0.005, *** *p* ≤ 0.001 vs. non-treatment basal, # *p* ≤ 0.05, ## *p* ≤ 0.005, ### *p* ≤ 0.001 vs. control sh-Nt cells with the same treatment (ANOVA-Bonferroni). Astrocytes primary cultures were infected with lentiviral particles containing a shRNA sequence against luciferase (control, sh-Ctl) or with shRNA targeting C/EBPβ (sh-C/EBPβ). Cultures were treated with 10 µg/mL of bacterial lipopolysaccharides (LPS) for 24 h and (**c**) immunoblot analysis were performed to determine C/EBPβ, TFAM, and ATPB protein levels. Data on the graphs correspond to the mean of the relative levels of these proteins normalized by α-tubulin in four independent experiments ± SE. * *p* ≤ 0.05, ** *p* ≤ 0.005, *** *p* ≤ 0.001 vs. non-treatment basal, ## *p* ≤ 0.005, ### *p* ≤ 0.001 vs. control sh-Ctl cells with the same treatment (ANOVA-Bonferroni). (**d**) The expression of TFAM and TOM20 was evaluated in these cells by immunofluorescence technique and representative images are shown. Scale bar = 20 µM.

**Figure 4 ijms-24-01459-f004:**
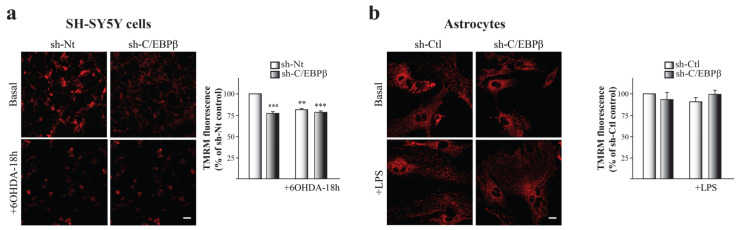
C/EBPβ knock-down causes mitochondrial depolarization in SH-SY5Y neurons but not in astrocytes. Mitochondrial membrane potential (ΔΨm) was estimated by live-cell imaging using TMRM in a redistribution mode (40 nM) in (**a**) SH-SY5Y cultures infected with lentiviral particles containing a Non-Targeting shRNA (sh-Nt) or with sh-C/EBPβ, which were treated with 35 µM 6OHDA for 18 h, and in (**b**) astrocytes primary cultures infected with lentiviral particles containing a shRNA against luciferase (sh-Ctl) or with sh-C/EBPβ, which were treated with 10 µg/mL of LPS for 24 h. Representative TMRM fluorescence images are shown and data are presented as the mean ± SE of TMRM fluorescence. TMRM fluorescence in control sh-Nt cells (**a**) or control sh-Ctl cells (**b**) was set as 100% and the rest of cellular conditions were referred to as percentage of the control cells measured in five different counting fields from at least three independent experiments. ** *p* ≤ 0.005, *** *p* ≤ 0.001 (ANOVA-Bonferroni). Scale bar = 20 μM.

**Figure 5 ijms-24-01459-f005:**
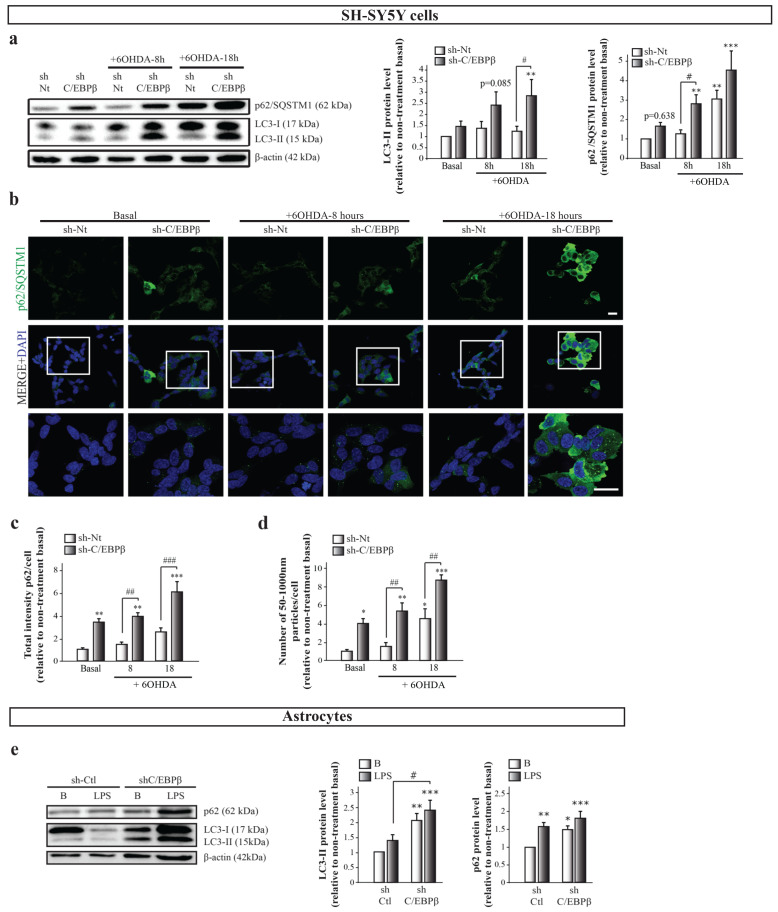
C/EBPβ silencing results in the accumulation of autophagy markers in cellular models of PD. SH-SY5Y cell cultures were infected with lentiviral particles containing a control sh-Nt or with a shRNA targeting C/EBPβ (sh-C/EBPβ). Cultures were treated with 35µM 6OHDA for 8 and 18 h and (**a**) immunoblot analyses were performed to determine LC3-II and p62/SQSTM1 protein levels. Data on the graphs correspond to the mean of the relative levels of these proteins normalized by β-actin in at least four independent experiments ± SE. (**b**) Representative images of p62/SQSTM1 (green) immunofluorescence analyses in the same cells with DAPI stained nuclei. Scale bar = 20 µM. The bar graphs show (**c**) the mean of the p62/SQSTM1 fluorescence intensity per cell (relative to non-treatment basal) ± SE or (**d**) the number of p62-containing particles with a diameter among 50–1000 nM per cell ± SE, which were considered autophagic vesicles (AVs). Both measurements were performed in five counting fields of three independent experiments. (**e**) Immunoblot analysis to determine LC3-II and p62/SQSTM1 protein levels in astrocytes primary cultures infected with lentiviral particles containing a control shRNA sequence (sh-Ctl) or with shRNA targeting C/EBPβ (sh-C/EBPβ), and treated with 10 µg/mL of bacterial lipopolysaccharides (LPS) for 24 h. Data on the graphs correspond to the mean of the relative levels of these proteins normalized by β-actin in four independent experiments ± SE. * *p* ≤ 0.05, ** *p* ≤ 0.005, *** *p* ≤ 0.001 vs. non-treatment basal, # *p* ≤ 0.05, ## *p* ≤ 0.005, ### *p* ≤ 0.001 vs. control (sh-Nt or sh-Ctl) cells with the same treatment (ANOVA-Bonferroni).

**Figure 6 ijms-24-01459-f006:**
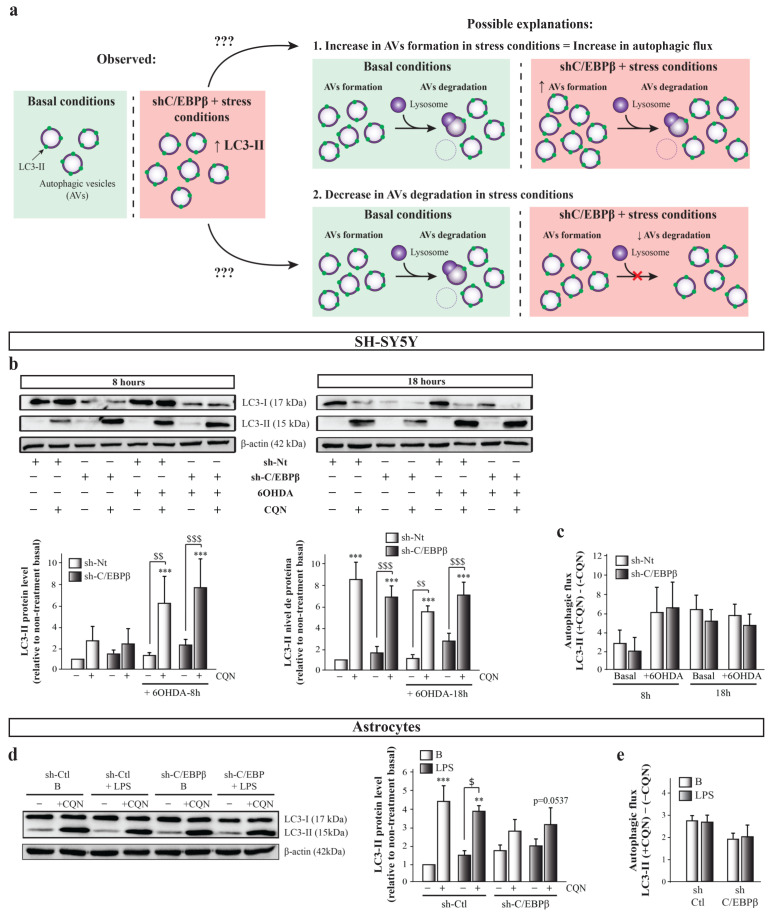
The accumulation of autophagy markers in C/EBPβ silenced cells in PD models is due to the alteration of autophagosomes degradation. (**a**) Scheme showing the possible explanations to the observed rise in autophagic markers levels. This can be due either to an increase in new autophagosome formation in PD cellular models in the absence of C/EBPβ, or to an alteration in the autophagy pathway that avoids autophagosomes degradation by lysosomes in these conditions. (**b**) To answer this question, autophagy was inhibited with chlorquine (CQN, 30 µM) in SH-SY5Y cells previously infected with lentiviral particles containing a control sh-Nt or with a shRNA targeting C/EBPβ (sh-C/EBPβ). Cultures were then treated with 35µM 6OHDA for 8 and 18 h and immunoblot analyses were performed to determine LC3-II protein levels. Data on the graphs correspond to the mean of the relative levels of LC3-II normalized by β-actin in at least three independent experiments ± SE. (**d**) Autophagy was inhibited with chlorquine (CQN, 30 µM) in astrocytes primary cultures infected with lentiviral particles containing a control shRNA sequence (sh-Ctl) or with shRNA targeting C/EBPβ (sh-C/EBPβ). Cultures were then treated with 10 µg/mL of bacterial lipopolysaccharides (LPS) for 24 h and immunoblot analysis were performed to determine LC3-II protein levels. Data on the graphs correspond to the mean of the relative levels of LC3-II normalized by α-actin in four independent experiments ± SE. Autophagic flux was calculated in (**c**) SH-SY5Y and (**e**) astrocytes cell cultures by subtracting the value of LC3-II protein levels in the presence of CQN (+CQN) minus the value of this levels in the absence of CQN (-CQN) in each experimental group. ** *p* ≤ 0.005, *** *p* ≤ 0.001 vs. non-treatment basal, $ *p* ≤ 0.05, $$ *p* ≤ 0.005, $$$ *p* ≤ 0.001 vs. the same experimental group without CQN (ANOVA-Bonferroni).

**Figure 7 ijms-24-01459-f007:**
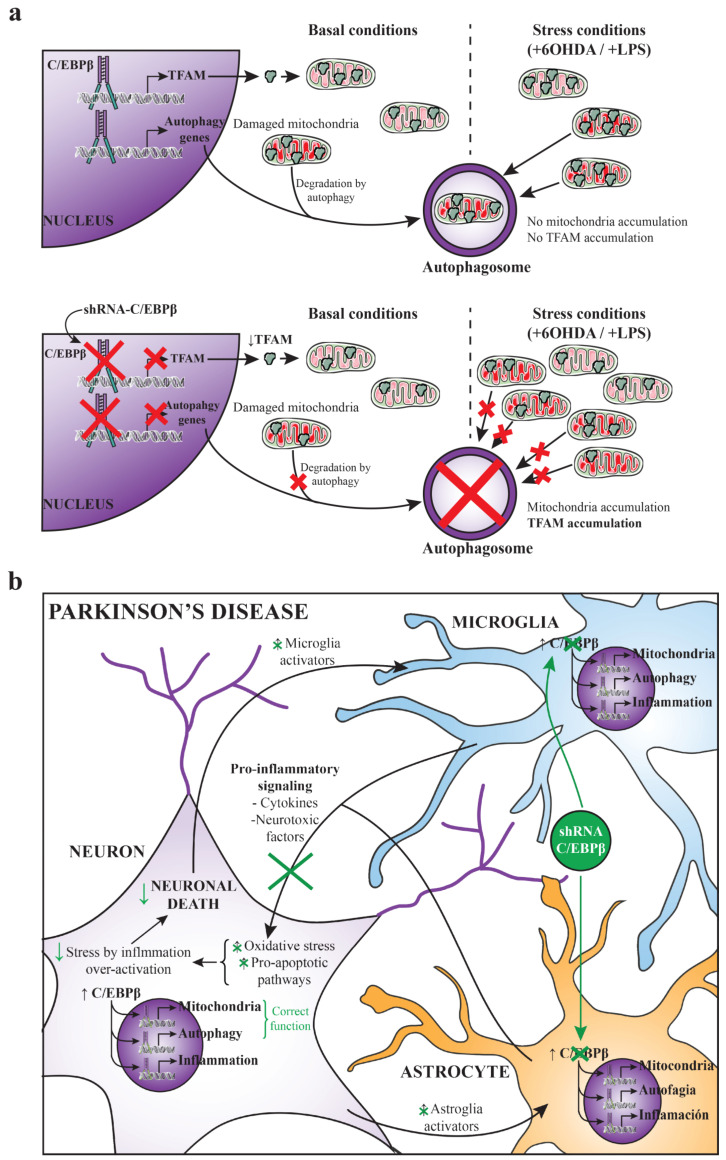
(**a**) Hypothesis of autophagy failure and mitochondria accumulation due to C/EBPβ silencing. With physiological levels of C/EBPβ, cells express TFAM and damaged mitochondria can be normally degraded by autophagy under basal and stress conditions produced by 6OHDA or LPS treatment. When C/EBPβ is silenced, TFAM expression regulated by this transcription factor is decreased; however, mitochondria degradation failure by the impairment of autophagy pathway could cause these organelles to accumulate and, consequently, the accumulation of TFAM. (**b**) C/EBPβ as a therapeutic target for the treatment of Parkinson’s disease. This scheme summarizes a possible strategy for the treatment of PD by silencing C/EBPβ specifically in astrocytes and microglia. Some of the events that take place in PD progression are shown in black. The beneficial effects of this approach that would increase neuronal survival are shown in green.

**Table 1 ijms-24-01459-t001:** Primers used for real-time quantitative PCR.

Gene Symbol	Forward Primer (5′ → 3′)	Reverse Primer (5′ → 3′)
*Hebp2*	CCGGCTCAAGTCCTTTCAGT	TCCATCAAAAGACCGCACGA
*Tfam*	CACCCAGATGCAAAAGTTTCA	CCACTCAGCTTTAAAATC CGC
*Slc25a37*	CAGCTATTGGTGCTTCTGGC	AAAGTGGGAACCTCCTCCCC
*18S*	CCAGTAAGTGCGGGTCATAAGC	CCTCACTAAACCATCCAATCGG

**Table 2 ijms-24-01459-t002:** Primers used for PCR amplification of TFAM promoter and deletions.

Sequence	Forward Primer (5′ → 3′)	Reverse Primer (5′ → 3′)
pTFAM/1188 (−1026/+162)	GGCTGTCTCAGAAGGTGGTTAG	CCACATGCTTCGGAGAAACG
pTFAM/652 (−490/+162)	TTAAGCTCTGCGGTAAGGCC	CCACATGCTTCGGAGAAACG
pTFAM/500 (−338/+162)	CTGGATGCAGGACTGTCTGT	CCACATGCTTCGGAGAAACG
pTFAM/352 (−190/+162	ACAGAGGTGGCTCAACAGC	CCACATGCTTCGGAGAAACG
pTFAM/270 (−108/+162)	TCTTATTCCTCCCCCGCAAG	CCACATGCTTCGGAGAAACG

## Data Availability

The data that support the findings of this study are available from the corresponding author upon reasonable request.

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
