# Peer review of "C/EBPβ Regulates TFAM Expression, Mitochondrial Function and Autophagy in Cellular Models of Parkinson’s Disease"

_ijms, 2023, doi:10.3390/ijms24021459_

Round 1

Reviewer 1 Report

The authors of this manuscript investigated the role of C/EBP-beta in transcriptional TFAM expression. This study was initiated from in silico model using C/EBP-beta knockout mouse brain and PD patient human brain samples. Although this study suggested an interesting role of C/EBP-beta in TFAM expression, several important issues should be intensively addressed.

1) Although the authors identified C/EBP-beta as a new transcription factor of TFAM, the role of C/EBP-beta mediated TFAM transcription in the PD model is unclear. In addition, although the authors conducted a luciferase assay using deletion mutants of the promoter region of TFAM, a ChIP assay should be performed to support their conclusion.  

2) In the second half of this manuscript, the authors focused on the role of C/EBP-beta in mitochondrial clearance (or quantity). However, several experiments should be improved. (a) To quantify mitochondria, more than checking the level of mitochondrial proteins, the authors had better check mtDNA contents. (b) The authors suggested that C/EBP-beta silencing causes inhibition of autophagic flux (by suppressing lysosomal activity). However, according to the results in Figure 6, autophagic flux tends to be increased by C/EBP-beta silencing and 6OHDA. In addition, since the authors want to know the mitophagy activity by C/EBP-beta, mitophagy activity should be measured using a mitophagy assay such as mito-Keima.

Specific concerns

1) In figure 1b, the authors identified that the mRNA level of TFAM is greatly decreased in the C/EBP-beta-KO brain. Since TFAM is an important factor in mitochondria biogenesis, it is hard to understand how mitochondria are accumulated even though the key mitochondria biogenesis factor (TFAM) is down-regulated.

2) The authors showed that the C/EBP-beta antibody detected both mouse and human forms in figure 2a. However, in Figure 2b, the authors showed the level of C/EBP-beta is only detected in p4 cells. Is this antibody specific for human form of C/EBP-beta?

3) The authors utilize primary astroglial cells (are those astrocytes? or glial cells?). To confirm whether most cells are glial cells, the authors should show staining images using glial marker antibodies.

4) When the authors show TMRE staining images, the authors should show mitochondrial markers together with TMRE to normalization.

Reviewer 2 Report

Parkinson's disease (PD) is a progressive neurodegenerative disorder, and despite its serious consequences, current treatments are only symptomatic and can only delay the degeneration of dopaminergic neurons. Therefore, a deeper understanding of molecular/cellular processes behind the pathophysiology of PD is critical in the research for new therapeutic targets. The current study is a consequence of the author's earlier findings, described in the paper by Morales-Garcia et al., 2017, where C/EBPβ mRNA expression and protein level were increased after 6-OHDA treatment, and C/EBPβ silencing attenuated the neuroinflammatory response and glial activation in 6-OHDA rat model of PD.

In the current paper, the authors discovered that: 1) C/EBPβ regulates mitochondrial genes; they focused primarily on mitochondrial transcription factor A (TFAM); 2) C/EBPβ activates the TFAM promoter activity; 3) C/EBPβ overexpression significantly increases TFAM levels during neuronal differentiation; 4) however C/EBPβ silencing down-regulates TFAM, it has no consequences on TFAM protein levels, because of mitochondrial accumulation; 5) in astrocytes, inflammation increases C/EBPβ protein levels, but during C/EBPβ silencing inflammation results in TFAM increase and mitochondrial accumulation; 6) C/EBPβ silencing decreases the degradation of autophagic vesicles (AVs) thereby leads to a dysfunctional mitochondrial accumulation. 

The results are well-presented, and most of them are thoroughly discussed. Hypotheses are clear; results are described in a logical sequence. A limitation is that they only present in vitro results in the paper, which makes the scientific significance of these results weaker. Addition of more in vivo data from the C/EBPβ KO animals would make the manuscript stronger. Also, some figures/pictures seem inconsistent with the graphs, and some results need to be discussed, or no conclusions were drawn from them. I suggest a revision based on the following comments:

- the authors did not describe how many mitochondrial genes they investigated with RT-qPCR. They found three down-regulated genes, but please provide information on why those three genes were chosen for PCR, and if there were more genes, please provide their PCR results in a supplement. Please also discuss in 2-3 sentences which disorders are the other two genes (Hepb2 and Slc25a37) involved. 

- If the authors investigate the effects of C/EBPβ KO on mitochondrial gene expression in relation to PD, what is the justification for examining hippocampal samples? 

- Sample size: authors used 10 WT and 5 KO animals for PCR. Why they did not increase the number of KO samples? If possible please add more KO samples and re-analyze the results. 

- The authors described an increase in TFAM levels due to C/EBPβ overexpression during neuronal differentiation. What do we know about other mitochondrial proteins? What do we know about mitochondrial accumulation? It would worth to investigate levels of other mitochondrial markers in this setting. 

- protein level bar graphs on Figures 2a, 3, 5, 6: What does "relative to basal" mean? It seems that basal is the control at timepoint 0. Please clarify the methods, how where these values are calculated, and what exactly "basal" means.

- In SH-SY5Y dopaminergic cells, treated with 6OHDA, TFAM and NDUFB8 protein levels decrease significantly after 4 hours, then return to their basal level. In the later experiments, this timepoint is missing, however, it would be interesting to see what happens with mitochondrial accumulation and autophagy in this timeframe. 

- some immunofluorescent and immunoblot pictures are inconsistent with the results on bar graph. Please revise those to make them more consistent. Figure 3 is an example of this inconsistency. The bar graph shows no difference in basal TFAM levels between silenced and controls. But both the immunoblot and the immunofluorescent photo suggest an increased TFAM in the silenced cells. Same is Figure 5 for p62/SQSTM1, where immunoblot and photo suggest a difference at basal timepoint but the bar graph not. Please reconsider the interpretation of results and provide a p value on the bar graph like at 8h timepoint. 

- on Figure 6b, please translate the text to English on the 2nd bar graph. "nivel de proteina" to protein level 

Round 2

Reviewer 1 Report

The authors addressed all my concerns appropriately.

Reviewer 2 Report

Thanks for appropriately answering my questions and addressing my comments.

The supplemental material and the changes in the manuscript are all addressing my comments/concerns, in my opinion, there is no need for additional experiments.